# SNP Markers and Evaluation of Duplicate Holdings of *Brassica oleracea* in Two European Genebanks

**DOI:** 10.3390/plants9080925

**Published:** 2020-07-22

**Authors:** Anna E. Palmé, Jenny Hagenblad, Svein Øivind Solberg, Karolina Aloisi, Anna Artemyeva

**Affiliations:** 1Nordic Genetic Resource Centre, Smedjevägen 3, SE-230 53 Alnarp, Sweden; anna.palme@nordgen.org (A.E.P.); karolina.aloisi@nordgen.org (K.A.); 2Department of Physics, Chemistry and Biology, Linköping University, SE-581 83 Linköping, Sweden; jenny.hagenblad@liu.se; 3Faculty of Applied Ecology, Agricultural Sciences and Biotechnology, Inland Norway University of Applied Sciences, NO-2418 Elverum, Norway; 4N. I. Vavilov Institute of Plant Genetic Resources (VIR), 42-44, B. Morskaya Street, 190000 St. Petersburg, Russia; akme11@yandex.ru

**Keywords:** *Brassica oleracea*, conservation, diversity, genebank, plant genetic resources, SNP

## Abstract

Around the world, there are more than 1500 genebanks storing plant genetic resources to be used in breeding and research. Such resources are essential for future food security, but many genebanks experience backlogs in their conservation work, often combined with limited budgets. Therefore, avoiding duplicate holdings is on the agenda. A process of coordination has started, aiming at sharing the responsibility of maintaining the unique accessions while allowing access according to the international treaty for plant genetic resources. Identifying duplicate holdings based on passport data has been one component of this. In the past, and especially in vegetables, different selections within the same varieties were common and the naming practices of cultivars/selections were flexible. Here, we examined 10 accession pairs/groups of cabbage (*Brassica oleracea* var. *capitata*) with similar names maintained in the Russian and Nordic genebanks. The accessions were analyzed for 11 morphological traits and with a SNP (Single Nucleotide Polymorphism) array developed for *B. napus*. Both proved to be useful tools for understanding the genetic structure among the accessions and for identifying duplicates, and a subset of 500 SNP markers are suggested for future *Brassica oleracea* genetic characterization. Within five out of 10 pairs/groups, we detected clear genetic differences among the accessions, and three of these were confirmed by significant differences in one or several morphological traits. In one case, a white cabbage and a red cabbage had similar accession names. The study highlights the necessity to be careful when identifying duplicate accessions based solely on the name, especially in older cross-pollinated species such as cabbage.

## 1. Introduction

A report from the Food and Agriculture Organization of the United Nations (FAO) indicates that up to 70% of the 7.4 million accessions around the world might be duplicate holdings [1]. At the same time, genebanks are struggling with inadequate resources and backlogs in regeneration, characterization, and documentation [2]. Taking a bird′s-eye view, duplication is not an efficient conservation approach. At a local level, each collection holder aims to have a large and influential collection. Requesting and maintaining accessions from other genebanks (duplication) has been one way to do this. The European Genebank Integrated System (AEGIS) has managed to involve institutions in more than 30 countries in an action for coordination and collaboration on plant genetic resource conservation [3]. The main idea is to share responsibility by establishing and operating a European Collection of unique and important germplasm and to increase the conservation efficiency and quality while facilitating the use of the genetic resources [4].

A critical step in this initiative has been the selection of accessions (generally seeds, conserved in genebanks). There are several challenges in such selections, but one is how to handle accessions with the same or similar names [5]. Same or similar names could be due to the duplication of accessions among genebanks, but can also indicate different selections (enterprises’ selections) and/or a flexible naming practice of seed material in the past [6,7]. Official variety lists (cultivar lists) and control came in the mid-twentieth century [8,9], and plant breeders’ rights came after the ratification of the International Union for the Protection of New Varieties of Plants, which was launched in 1969 [10]. In 1969, local companies’ selections were still listed under similar names in Scandinavia, but almost all of them were removed from the national variety lists between 1970 and 1980 [11]. All this has resulted in a large number of older varieties with the same or similar names. An important question is whether accessions of such varieties should be regarded as duplicates when efforts are made to increase the efficiency in genebank conservation. 

Recent developments such as using passport data with digital object identifiers (DOI) on accessions and transactions [12], large-scale morphological characterization and phenotyping [13], and molecular studies with next-generation sequencing platforms [14,15,16,17,18] have all improved the possibilities to identify duplicates. Regardless of the approach, proper data and transparent genebank information systems are needed to facilitate duplication assessments [19,20]. *Brassica oleracea* L. (2n = 18) comprises many important crops, including cauliflower, broccoli, and cabbages as well as wild species and subspecies which are cross-compatible with the cultivars [21,22]. The issue of duplicate genebank holdings of *B. oleracea* has been raised [23], and genetic diversity has been investigated using, for example, AFLP (Amplified Fragment Length Polymorphism) markers [24,25,26] and microsatellites [27]. In these studies, substantial diversity within accessions was observed, but so was a clear differentiation among accessions. A high-density SNP genotyping array for *Brassica napus* and its ancestral diploid species has been developed [28], and in this study we test it on *B. oleracea*. 

The main objectives of this study were: (1) to examine the suitability of this Illumina Infinium SNP array to study the genetic diversity and structure in cabbage; (2) to evaluate potential duplicates among genebank accessions with similar names by using morphological traits and the mentioned genetic markers. If the SNP array could be used successfully, we wanted to identify a sub-set of SNP markers to be used for the future screenings of a larger number of accessions in a process of identifying duplicates and incorrectly labelled accessions at a European or global scale.

## 2. Results

The *Brassica* Working Group of the European Cooperative Programme for Plant Genetic Resources (ECPGR) has prioritized AEGIS, which means there is an ongoing process of searching for potential duplicates. Based on the cabbage passport data from the N. I. Vavilov Institute of Plant Genetic Resources in St. Petersburg (VIR) and the Nordic Genetic Resource Centre (NGB), we were able to identify 40 pairs, triplets, and groups (hereafter termed groups) based on the “accession name” or “donor name”. Here, we present the results for 10 such groups, with a total of 27 accessions (Table 1).

### 2.1. Morphological Diversity

Two types of morphological descriptors were analyzed: continuous descriptors, such as plant height and leaf length, and categorical descriptors, such as leaf color and head shape. Many of the continuous, numeric descriptors were positively correlated (arrows pointing to the right in Figure 1). The two first principal components represented 42% and 23% of the total variation. Accessions with the same or similar names grouped together, but only to a certain extent.

Within the Amager Tall group, A1 (Amager Hög) and A4 (Grami) did not cluster with A3 (Amager Høj Grøn Grami), A6 (Amager Høj, Grøn, Toftø 67), and A7 (Amager Tall Resistent). The biplot furthermore indicated that the two Amager Winter accessions (A11 and A12), the two Stavanger Torg accessions (ST1 and ST2), and the two Kissendrup accessions (K1 and K2) did not cluster within their respective group.

The two Stavanger Torv accessions were early maturing and needed around 120 days to maturation, while most of the other accessions needed 140 days or more (Figure 2). There was also a large variation in the time to maturity and other traits within many of the accessions. The results from the Tukey multiple comparisons of means (Appendix A) showed that A1 differed significantly from A4, A6, and A7 in the time to maturity (all *p* < 0.05). Furthermore, A4 differed from A1 and A6 in the leaf lamina length and plant height, and A4 from A1 in head weight (all *p* < 0.05). A11 and A12 differed in plant height and, notably, also in leaf lamina color, where A11 plants were purple while A12 plants were green. Significant differences were also detected among the Langendijker Summer accessions. The L2 and L3 plants were purple, while L1 had a mixture of purple and green leaves. In addition, L1 differed from L2 in core length (*p* < 0.05). The Stavanger Torg accessions ST1 and ST2 differed in head height and head density, while the Kissendrup accessions differed in the time to maturity (all *p* < 0.05). No clear differentiation was detected among the accessions within the Blåtopp, Ruhm von Enkhizen, Loke, or Jåtunsalgets Vinterkål groups (Appendix A).

### 2.2. Marker Efficiency and Accession Diversity

Among the 5965 markers, 3969 failed to amplify in all the genotyped cabbage plants. Of these, mapping data was available for 3750 markers, 99% of which were located on the A genome in *B. napus*, and therefore were not expected to be found in *B. olereacea*. The largest proportion of failed markers was found on the *B. napus* chromosome A04 (0.540), with the lowest on chromosome C06 (0.002). Duplicate samples showed a high consistency across runs. Each re-genotyped individual differed in only two markers. Individual 135 differed in Bn-A01-p16331424 and Bn-scaff_15712_6-p1025930, and individual 136 in Bn-scaff_15877_1-p926737 and Bn-scaff_16553_1-p6743.

The highest average number of alleles across all the loci was found in accession A4 (1.8) and the lowest in accession K1 (1.2) (Table 2). The same accessions had the highest and the lowest observed heterozygosity and the lowest genetic diversity, calculated as Nei’s h (expected heterozygosity under Hardy–Weinberg Equilibrium, HWE) (Table 2). In some cases, accessions with similar names had similar levels of within-accession diversity—for example, ST1 and ST2—but in many cases, different levels of diversity were observed among the accessions within a group (Table 2). The genetic diversity of the accessions from NGB did not differ significantly from those from VIR (*t*-test, *p* = 0.734). Genetic diversity was not significantly correlated with acquisition year for the full data (*p* = 0.177) nor for accessions from NGB (*p* = 0.687), but was positively correlated with the acquisition year for VIR accessions (c = 0.553, *p* < 0.05).

### 2.3. Accession Comparisons

All the pairwise F_ST_ values were significantly different from 0 (*p* < 0.001 for all comparisons), indicating genetic differentiation among all accessions. In general, the average pairwise F_ST_ values were lower between pairs of accessions belonging to the same group (average 0.193) than between pairs of accessions belonging to different groups (average 0.270) (*t*-test, *p* < 0.001). The lowest F_ST_ value (0.087) was found between accessions B6 and B3 (Table 3), both from the Blåtopp group and with a similar morphology (Figure 1, Appendix A). The same was true for the Loke pair (LO1 and LO2), with a very low F_ST_ between the accessions (Table 3) and no significant morphological differences (Appendix A). The highest F_ST_ values within groups were found between the pairs A8 and A9 (0.326) and B2 and B3 (0.321, Table 3), accessions showing little morphological differentiation (Figure 1, Appendix A). The highest F_ST_ value overall was found between the accessions A1 and K1 (0.564, Appendix A), the two accessions with the lowest level of genetic diversity. The pattern was reflected when looking at the average number of pairwise shared alleles, where accessions A1 and K1 shared few alleles with many of the other accessions. The lowest number of shared alleles was found between the accessions K1 and A11, and the highest between A1 and A4 (Appendix A).

A STRUCTURE analysis showed equally high support for two and three clusters (Appendix A). At K = 2, the first cluster contained A1, A3, A6, A7, A12, B1, B3, B6, J1, and J2 and the second contained A8, K1, K2, L1, L2, and L3. The remaining accessions showed some level mixed clustering, with similar degrees of mixture for R1, R2, R3 and ST1 and ST2 (Figure 3). At K = 3, the cluster consisting of accessions A8, K1, K2, L1, L2, and L3 remained intact. A second cluster consisted of some of the accessions showing mixed clustering at K = 2: R1, R2, R3, and ST1. The third cluster contained accessions A1, A3, A6, A7, A12, B1, B3, B6, J1, and J2, with the remaining accessions showing various degrees of mixed clustering (Figure 3b). In general, accessions within the same group tended to belong to the same clusters (Figure 3). For example, all the Kissendrup, Langendijk, Ruhm von Enkhuizen, Jatunsalgets, and Loke accessions clustered within the same group. The most notable exception was accession A8, Amager Kurzstrunkiger Original. Based on the accession name, A8 was expected to cluster with the other Amager accessions, but instead the accession clearly clustered with the Kissendrup and Langendijker accessions (Figure 3).

The PCA based on the allele frequencies in the different accessions (Figure 4) supported the clustering found with STRUCTURE (Figure 3) to a large degree and the morphology-based clustering to a lesser degree (Figure 1). One cluster contained A8, K1, K2, L1, L2, and L3, (upper right); one cluster contained accessions R1, R2, R3, and ST1 (lower center); and one cluster contained accessions A1, A3, A6, A7, A9, A12, B1, B3, B6, J1, LO1, J2, and LO2 (upper left). Accessions A4, B2, A11, and ST2 were located between the latter two clusters, substantiating the structure analysis at K = 3. There was no evidence of clustering according to the genebank origin. The VIR accessions acquired at an earlier date tended to be located less centrally in the PCA (c = −0.540, *p* = 0.056) than the NGB accessions. An individual-based PCA showed a good agreement with the accession average-based PCA. Most individuals of each accession clustered together, but two exceptions were found in accession L3 and accession A4 (Appendix A).

### 2.4. Genotypic and Morphological Comparisons

In the accession-level PCA based on the SNP data, the first principal component (PC1) separated most of the purple-leafed accessions from most of the green-leafed accessions. The green-leafed accession A8, however, clustered among the purple-leafed accessions, while the purple-leafed accession A11 clustered among the green-leafed (Figure 4). The accession L1, with purple/green leaves, clustered among the purple accessions. Neither head density nor head shape showed any clustering in the accession-level PCA (data not shown).

### 2.5. Limiting the Number of SNP Markers

Subsets of the data were analyzed to determine whether a more limited number of markers could be used to capture a similar amount of information as the full dataset. Already, with as few as 10 randomly chosen markers, the F_ST_ values obtained were in most cases similar to those calculated from the full dataset (Table 3). Increasing the number of markers to 50, 100, and 500 markers, respectively, reduced the variance of the F_ST_ estimates (Appendix A) from an average of 0.088 for 10 markers to 0.011 for 500 markers. The mean F_ST_ values did, however, not change consistently in any given direction, and could either increase or decrease with an increasing number of markers. The mean F_ST_, however, always changed less than 0.01.

Individual-level PCA clustering according to accession could be obtained with a limited number of markers. Already, when subsampling the 20 markers with the alleles providing the largest segregation along PC1 and PC2, a reasonable clustering according to accession could be obtained (Appendix A). The same number of markers used on the accession-level PCA could only separate accessions along PC1 (Appendix A).

A single subsample for each 10, 50, 100, and 500 markers was used to investigate whether the same clustering could be obtained in a STRUCTURE analysis as with the full dataset. All subsets indicated K = 2 to be the level of structuring best explaining the data. With only 10 markers, very limited power was obtained to identify structuring, although the clustering of the Jatunsalgets Vinterkål group (J1 and J2) and the Ruhm group (R1, R2, and R3) could be discerned. Surprisingly, as few as 50 markers yielded a structuring similar to that obtained for the full dataset, as well as from 500 markers (Appendix A). A STRUCTURE analysis of 100 randomly chosen markers, however, showed that a lower number of markers was not sufficient to reliably replicate the results of the full dataset.

To explore the efficiency of 50 markers to capture the same structure at K = 2 as the full dataset, an additional 9 datasets of 50 randomly chosen markers were generated. Comparisons of the STRUCTURE analysis of the 10 sets of 50 markers showed that some but not all clusters were reliably identified in each subset (Appendix A). In particular, the clustering of the Jatunsalgets Vinterkål (J1 and J2) and the Loke (LO1 and LO2) groups varied. An analysis with the software CLUMPP showed that the clustering among the 10 sets was not very consistent (H = 0.752).

A subset of markers was chosen with the aim to provide a good resolution for discriminating between accessions. The markers were chosen to provide as high a discriminatory power as possible along the first two principal components in the accession-level PCA (Figure 4). The markers were further evaluated to show a high level of genetic diversity (h > 0.3) and not found to be in high linkage disequilibrium (D′ < 0.25) with each other. In total, 500 markers were chosen (Appendix A).

## 3. Discussion

### 3.1. Discrimination Power and the Number of Markers

The array used in this study for genotyping *B. oleracea* was originally developed for *B. napus* [28]. Not surprisingly, the array showed the greatest efficiency for markers located on the “C” genome in *B. napus*, with less than 1% of the markers on most chromosomes failing to amplify. Nevertheless, as many as 50% of the markers mapping to the “A” genome in *B. napus* were able to successfully amplify in our *B. oleracea*. Further studies are needed to discern the mapping location of these markers in *B. oleracea* and to evaluate the amount of cross-amplification between the “A” and the “C” genome.

Although the cost of genotyping and sequencing is becoming ever lower, using a limited number of markers while retaining sufficient discriminatory power is still of interest. We found that a random sample of only a few percent of the amplifying markers would have provided us with an overall picture of reasonable similarity to the one obtained with the full dataset. In most cases, the F_ST_ value between two accessions in the same group could be estimated with reasonable accuracy with as few as 50 or 100 markers, similar to what was reported by Willing et al. [29].

A STRUCTURE analysis further suggested that as few as 50 randomly chosen markers could often capture a large part of the genetic structuring in the data, although the clustering of some accessions was inconsistent and the single analysis of 100 random markers failed to replicate the clustering obtained with the full dataset. In addition, parameters such as the amount of gene flow and the evenness of sampling have been shown to influence the number of markers needed to discriminate between groups of differently related individuals [30]. The minimum number of markers needed will vary from organism to organism and from case to case, but may be surprisingly low. Choosing the 20 most discriminatory markers for individual-level PCA resulted in an outcome with a high similarity to that of the full dataset. However, these 20 markers might not be the most discriminatory in another sample of cabbage accessions. For this reason, we recommend a subset of at least 500 SNP markers for duplication assessments in cabbage in order to get a robust, detailed result. A list of 500 SNP markers with a high discriminating power, high level of genetic diversity, and low linkage disequilibrium with each other can be found in Appendix A).

### 3.2. Duplication Assessment and Genetic Similarity

This study clearly demonstrates that the same or similar names do not necessarily mean a duplicate holding. Both the STRUCTURE and PCA analyses detected genetic differences among accessions grouped by accession name. In the STRUCTURE analysis, five out of the 10 groups showed genetic differences within the group: A4 vs. A1, A3, A6, and A7 (Amager Tall group); A8 vs. A9 (Amager Short group); A11 vs. A12 (Amager Winter group); B2 vs. B1, B3, and B6 (Blåtopp group); and ST1 vs. ST2 (Stavanger Torv group, Figure 3). A similar pattern could be seen in the PCA based on SNP markers (Figure 4). Three of the five cases were supported by significant differences in one or several morphological traits (Appendix A). The difference between the two Amager Winter accessions was obvious from the morphology, where A11 (Amager Winter) was identified as a red cabbage type, while A12 (Amager Vinter Gefion) was a white cabbage. The differences between the two Stavanger Torv accessions (ST1 and ST2) were less obvious, both being early maturing white cabbage types but with significant differences in two head characters. Within the Amager Tall group, morphological data also confirmed that accessions were different (Appendix A), while no clear differences in morphology were observed in the Amager Short group or the Blåtopp group.

Genetic similarity can be used as a criterion to identify and handle duplicate holdings in genebanks [23,24]. In clonal and highly inbred material, it can be a relatively easy task to determine whether accessions are duplicates, but in open pollinated crops such as cabbage, the genetic structure is more complex. Even accessions that have the same origin—for example, accessions donated from the same seed lot to different genebanks—are expected to diverge over time [27]. In these species, the overall pattern of genetic diversity needs to be taken into consideration.

In instances where several lines of evidence (e.g., low F_ST_ values, a large proportion of shared alleles, morphologic similarity, and common clustering in the STRUCTURE analysis and PCA) suggest a high similarity compared to the average, accessions can be considered duplicates and one of the accessions can be removed from the genebank holdings with a minimum loss of genetic diversity (or bulked) [23]. Examples from our dataset that could be bulked, removed, or given lower priority in the conservation could be B3 and B6 and one of the LO accessions. Our study has shown that using accession names alone is not a good strategy to reduce duplicate holdings, as the same or similar names does not mean identical genetic composition. A combined method using both accession names and other passport data as a first step and then marker evaluations as a second step would be a better approach. Alternatively, morphological evaluations could be used or a more extensive passport data evaluation trying to trace the transactions of accessions between genebanks, for example, by using donor accession numbers or other relevant information. The ECPGR Brassica group has established an online tool for identifying duplicate holdings based on accession names and other passport data. This is a useful first step that could be taken into a next step with an extensive evaluation of the potential duplicates with the developed marker set.

### 3.3. Cultivation History and Naming Practices

There can be several explanations as to why accessions with the same or similar names are genetically and/or morphologically different. Minor differences could be explained by breeding history and naming practices. As mentioned, selections within a cultivar were common in the 19th and first part of the 20th century, and a cultivar could have many and complicated names [7]. One example is “Jatunsalgets Vinterkål Berbes St. Orginal” (J1) or “Jåtunsalgets Vinterkål” (J2). These two accessions are morphologically close and cluster together both in the STRUCTURE analysis and the SNP-based PCA, but the accessions are not identical, either morphologically or genetically. Jåtunsalget was a small Norwegian seed enterprise with only these two accessions recorded in the ECPGR *Brassica* database [31]. “Jåtunsalgets Vinterkål” was listed on the Norwegian variety list in 1979 [32], however the pedigree was “Jåtun Amager x en Hollansk sort i 1929”, which means “a selection of Amager crossed with a Dutch cultivar in the year 1929”. Most likely, “Jåtun Vinterkål” was marketed already in the 1930s but was listed much later. The oldest (and most original) accession is J1, acquired by VIR in 1953, while J2 (from NGB) entered the Nordic Genebank more than 20 years later from unknown sources [32].

A more complicated example is the Amager varieties. The ECPGR *Brassica* database [31] shows 102 records with “Amager” in their accession name. Amager is a geographical area and a village just outside Copenhagen that hosted both seed enterprises and an extensive vegetable production. We divided the Amager accessions in our study into three sub-groups based on naming; one of them was the Amager Tall sub-group, where the Danish word “Høj” or the Swedish “Hög” both mean “Tall” (with 21 accessions in the ECPGR database). Our study included five such accessions where A4 (accession name “Grami”) was genetically clearly different from the other four. In most selections, there is a second name—e.g., “Grön Grami”, “Grön Toftø”, or “Resistent”—describing further selection properties or enterprises’ names. We hypothesized that A4 (“Grami” from VIR) would be similar to A3 (“Amager Høj Grøn Grami” from NGB), but this was not the case. In retrospect, we should maybe not have grouped A4 with the Amager Tall accessions, as the only link to Amager was through the name “Grami”, which was used also in the name of A3.

Within the remaining two Amager sub-groups, genetic differences were also detected. Amager Winter is commonly known to be a white cabbage (green leaf laminas), marketed, amongst others, by A. Hansen Amagerfrø in Denmark in the mid-20th century [7]. In the ECPGR *Brassica* database [31], there are six accessions fitting this name, and we included two of these. Test cultivations showed that A11 (Amager Winter, K192) was a red cabbage (purple leaf laminas). A11 is from an unknown source in Denmark, acquired by VIR in 1969, and the accession has so far been through at least six regeneration cycles at the VIR experimental field. The accession was listed as purple at the time of entry (as accession number K192 in the VIR catalogue). Certainly, there have been red Amager cabbages traded. According to the Nordic Genetic Resource Center cultivar database [32], a red cabbage cultivar/selection named “Amager” was released in 1959 and was bred by Østergård frøavl (Denmark, breeders name Stenballe P 59 68). Other red Amager cabbage cultivars/selections were “Amager 304”, bred by A. Hansen Amagerfrø (breeders name Tagenhus P 59 69, released in 1959); “Holdbar Amager”, bred by L. Dæhnfeldt (breeders name Toftø S 1960, released in 1960); and “Amager Caro” and “Amager Rega”, both released in 1974 by Ohsens Enke (Denmark). A11 could not be one of the latter, as it was acquired by VIR already in 1969, but it could be one of the earlier developed red “Amager” cultivars. What is certain is that A11 is not a duplicate of A12, which has a similar name (Amager Vinter Gefion, NGB1879) but is a genetically different white cabbage. Although A11 is different from the remaining accessions in the “Amager” group in the PCA, it does show a higher similarity to the “Amager” group than to other red cabbages in the PCA (Figure 4). This, together with the results of the STRUCTURE analysis (Figure 3), tentatively suggests that the accession is the result of breeding the purple color into an Amager background. It is hard to know if the polymorphism observed in accession L1 (compared to L2 and L3; Figure 3) is due to gene flow from accessions with a different leaf color, as there are both green and purple plants in this accession, or if there are other explanations.

The genetic data showed clear differences between the Amager Shor pair (Amager Kurzstunkiger Orginal, A8) and (Amager L NF Orginal, A9). Based on passport data, we know they are from different seed companies (one in Denmark and one in Norway) and were included in the collection at VIR in different years (1935 and 1967, respectively). Accessions with the name “Amager Kurztunkig” are found in in Germany, Poland, the Check Republic, and Belarus [31], most likely duplicated with the original accession from VIR included in this study. The prefix “Kurz” is a German word and means “Lav” (in Scandinavian) or short in English. Our reason for pairing A8 and A9 was the prefix “Kurz” in A8 and the abbreviation L (“Lav”?) in A9. Regarding plant height, they were both short, and they had quite similar morphological characteristics, but were not very close in the morphological PCA (Figure 1). Most likely, A9 is an Amager selection from Norsk Frø (NF). From Norway, a cultivar with a similar name (Amager L1 Orginal, not included in this study) is known, with the pedigree “Jåtun Amager x Jåtunsalgets vinterkål 1932” [32]. The cultivar was marketed from 1933 onward, but it was approved as late as in 1961. Amager Kurzstunkiger came to VIR in 1935 from a Danish enterprise but with a German accession name. Certainly, A8 and A9 have a different history and, as demonstrated, they are genetically different.

### 3.4. The Effect of Genebank Conservation

Changes in genetic composition, major or minor, may take place during field regeneration [33,34,35,36,37]. Our study was not designed to track changes from generation to generation, but some of the differences we observe are probably the result of this process. Minor changes are expected during regeneration in genebanks, especially if a low number of plants are used [38] or if insufficient isolation is used during flowering. 

Genebanks use a standard number of plants, usually 20 to 50 individuals, and net cage isolation with pollinators to reduce the risks of genetic changes during regeneration. The FAO genebank standards do not specify the number of plants [39]. At VIR, regeneration takes place typically every 5–7 years, meaning that material acquired in the 1930s has been through at least 10 regeneration cycles. This is expected to result in an increase in differentiation among accessions and a loss of genetic diversity within accessions. The heterozygosity is predicted to decrease each generation in proportion to the population size [40]. For example, with 10 regeneration cycles and a population size of 20, a 22% decrease in heterozygosity is expected on average (H_t_ = (1−1/2N) H_t_−1; N=diploid population size). We found that accessions acquired a long time ago tended to have lower genetic diversity than more recent additions, a pattern that is in agreement with the loss of genetic diversity from genetic drift. Additionally, more recently acquired accessions tended to cluster more centrally in the PCA plots, which could also be the result of genetic drift acting to differentiate older accessions.

Other factors such as selection and gene flow could also affect genebank accessions. Some selection from local conditions—both those connected to the environment at the regeneration site, such as climate and soil conditions, and those linked to cultivation practices, such as harvesting time and methods—is expected. In addition, if accessions are not completely isolated, gene flow will occur from other cultivated genebank accessions of the same species, from cultivated fields in the area, and from weeds. Bees and flies are the main pollinators of cabbages [41], and to avoid unwanted gene flow through cross-accession pollination, isolation is crucial. Contrary to genetic drift, external gene flow is expected to increase diversity within the accession and decrease divergence among accessions cultivated at the same time.

Van Hintum et al. [24] demonstrated that the genetic changes caused by regeneration within an accession were of similar magnitude to differences among genebank accessions of cabbages with the same or similar names. Therefore, they questioned the rationale behind conserving a large number of accessions with the same or similar names. Our study supports the occurrence of genetic change during regeneration and similarity among some accessions with similar names. At the same time, however, we have shown that similar names do not always imply the same genetic material.

### 3.5. Implications for Genebank Conservation

All genebanks have limited budgets and need to adapt their operations to their economic frame. One way to adapt is to remove or pool/bulk duplicates and thus make funds available for the high-quality conservation of the remaining unique accessions. This approach has been used in many species, including *Brassica* crops [23,42].

AEGIS has suggested a roadmap for how to handle duplicate holdings at the European level, identifying the most appropriate accessions based on passport data [4]. However, the decision to remove duplicates is the responsibility of individual genebanks. Our study clearly shows that relying exclusively on accession name when identifying duplicates can be risky, especially with old cross-pollinating cultivars with complex breeding history and naming practices. We find that in five out of the 10 groups, accessions with same or similar names have clear genetic differences. In most cases, such differences were corroborated by significant differences in one or several morphological traits.

Additional passport data such as accession numbers, donor institute, donor accession number, etc. can help pinpoint the origin of the accession and the time of split from other accessions. If such data is available, the chance of correctly identifying duplicates based solely on documentation increases. For recently acquired material, for example—modern cultivars—this can be a safe approach. However, for older cultivars and landraces this is often difficult. Documentation is often missing and, as discussed above, accessions can have diverged substantially from a common origin via complex breeding histories and regeneration in genebanks.

Using detailed morphological characterization has been suggested [43], as has combined morphological and molecular characterization, and molecular characterization alone [18,44]. Our current findings lend support to the need for characterization before deciding to remove or bulk accessions. However, a major challenge is the costs of such characterizations. Most genebanks are underfunded and have backlogs in regeneration and viability monitoring. For tracking future duplications, the introduction of DOI on the accession level [12] would make transactions between genebanks easier, but it cannot capture what is already duplicated. International collaboration and genotyping using the next-generation sequencing could be a cost-effective way forward [14,15,16,17,18]. Here, proper information on the accession level could go hand-in-hand with the facilitation of the use of the germplasm as genetic information is catalogued, linked to the accessions, and made available for the users.

## 4. Materials and Methods

### 4.1. Plant Material, Cultivation, and Morphological Characterization

In Europe, there are 35 *Brassica* collections, located in 24 countries and with more than 11,000 *B. oleracea* accessions [45]. In total, 980 cabbage accessions are maintained at VIR and 189 at NGB. An initial study [46] characterized six groups for morphological traits, and some differences within the groups could be detected. In this study, another 10 groups (Table 1) were examined to see if there were differences among accessions within a group. For each group 10 plants per accession were planted. These plants were randomized, and each of the 10 plants was characterized. The study consisted of 10 such randomized pair/triplet characterizations. The planting distance was 50 cm between the plants. The work was done at Alnarp, Sweden (55°’ N, 13°’ E); the soil was loamy clay and the fertilization was 100 kg ha-1 PROMAGNA 11-5-18^™^ (Yara, Norway) at planting and 30 kg ha-1 YaraMila 22-0-12^™^ (Yara, Norway) one month after planting. Plants were irrigated and biological control and fungicides were applied. Plants were evaluated just before harvesting. SI units were used for plant leaf, head, and core size parameters and UPOV (International Union for the Protection of New Varieties of Plants) [47] descriptors for leaf color, head shape, and head density. Details are provided in Appendix A. Principal Component Analysis (PCA) bi-plots were used for an overview of the data and characters. An ANOVA was performed for each numeric character and included data from all the individuals in that group. If the ANOVA indicated significant differences among accessions, a Tukey multiple comparison of means [48] was used to identify accessions that differed from each other. χ2 statistics were used for categorical characters.

### 4.2. DNA Extraction

As far as we know, the selected accessions have not previously been included in any molecular studies. DNA extraction was conducted on the same 10 plants per accession that were morphologically characterized. Leaf samples (2 cm^2^) were collected from the plants cultivated in the field, placed in 2 ml Eppendorf tubes, immediately frozen in liquid nitrogen, and subsequently freeze-dried overnight in a LyoLab 3000 (Heto Lab Equipment). The freeze-dried material was powdered in a mixer mill (Merck Retsch MM 300) using steel beads; hereafter, 600 μl of CTAB buffer was added to each powdered sample (0.1 M Tris; pH 8.0, 0.01 M EDTA, 0.7 M NaCl, 1% CTAB, and 1% β-mercaptoethanol) according to Doyle and Doyle [49]. The samples were incubated in a thermomixer (Eppendorf) at 600 rpm and 60 °C for 60 min. DNA was extracted by adding one volume of chloroform/isoamylalcohol (24:1), then they were mixed and centrifuged for 20 min at 13,200 rpm. The supernatants were transferred to new tubes and 5 μl of RNAse (1 mg/ml) was added and incubated in the thermomixer (600 rpm, 37 °C for 30 min). Cold isopropanol (0.8 V) was added, followed by mixing and centrifugation (10 min at 13,200 rpm). The DNA pellet was cleaned in 500 μl of wash-buffer (76% ethanol, 0.2 M sodium acetate) for 20 min at room temperature with subsequent centrifugation for 5 min in 13,200 rpm, followed by 500 μl of rinse-buffer (76% ethanol, 0.01 M ammonium acetate) and mixing and centrifugation for 5 min at 13,200 rpm. The samples were left to dry in room temperature for 1 h and were then re-suspended in 50 μl of ddH_2_O. The DNA concentration and ratio (at 260 nm and 280 nm) were determined using an Eppendorf BioSpectrometer. The DNA concentrations of the samples were adjusted for further analysis.

### 4.3. SNP Analysis and Statistics

Array genotyping was performed by TraitGenetics GmbH (Gatersleben, Germany) and with a 15 K Illumina Infinium array that contains a subset of markers from the Brassica 60K array [28]. The array had previously been tested for *B. oleracea* and had a total of 13,714 SNPs. An initial run was carried out with 92 individuals, and a second with 178 individuals. Ten individuals were analyzed from each accession in order to capture the within-accession variation. Cabbage accessions are expected to harbour substantial within-accession variation, and therefore more individuals per accession are needed. By analyzing 10 individuals per accession, we gained an adequate picture of the within-accession variation and at the same time we were able to include many accessions in the study. The first run with 92 individuals was a test run, and since that was successful the second run with 178 individuals was performed in the same way to increase the number of individuals analyzed. After the merging of the two runs, failed and invariant markers were removed, as were markers failing in more than 50% of the individuals. The remaining 5965 markers were used for further analysis. Of these, 68% (4090 markers) had less than 10% missing data. After the removal of the above markers, individuals with more than 40% failed markers were removed (2 individuals + 2 controls). Of the remaining individuals, all had less than 10% missing data.

Two individuals of the accession K2248 were genotyped in both runs. The second genotyping of both individuals had a lower success rate and was removed from the downstream analysis. In total, 266 individuals were kept for further analyses and were analyzed with 5965 markers.

Deviations from the HWE (Hardy-Weinberg Equilibrium) were tested using a χ2 test with and without Bonferroni correction. All the accessions had less than 10% of the markers deviating from the HWE before the Bonferroni correction. No marker deviated significantly from the HWE in any accession after the Bonferroni correction, and hence no marker was removed for this reason.

Wright’s F_ST_ [50] and Nei’s h were estimated according to Nei [51] using purpose-written perl scripts. For the F_ST_ values, significance was determined by permutation tests (1000 permutations). Subsets of the markers were analyzed to evaluate if the F_ST_ values between pairs of accessions from the same group could have been estimated with similar accuracy with a more limited number of markers. Subsets of 500, 100, 50, and 10 markers were randomly drawn from the dataset and used for calculating the F_ST_ values. This was repeated 1000 times for each number of markers, and the average F_ST_ values and standard error for the 1000 replicates were calculated.

PCA of the genetic data was carried out using R v 3.2.4 [52] and the *prcomp* command. For an accession-level PCA, the allele frequencies for each allele at each locus were treated as independent variables, while in the individual-level PCA the number of copies of each allele at each locus was used.

The software STRUCTURE (v 2.3.4) [53,54] was used to explore the data for genetic structuring. The software was run with a burn-in length of 20,000 iterations, followed by 50,000 iterations for estimating the parameters, with non-amplifying markers treated as missing data. Each analysis using the admixture model was repeated 10 times for each number of clusters (K = 1 to 10), until the likelihood values for the runs no longer improved. The number of clusters observed in the dataset was evaluated by calculating the ΔK according to Evanno et al. [55]. CLUMPP v 1.1.1 [56] was used to compare the results of individual runs and to calculate the similarity coefficients, H, and the average matrix of ancestry. In CLUMPP, the Full- Search, Greedy, and LargeKGreedy algorithms were used for comparing runs with K < 4, 4 ≤ K ≤ 6, and K > 6, respectively. The graphical presentation of the results was obtained using DISTRUCT v 1.1 [57]. STRUCTURE was also used to analyze the subsets of randomly chosen markers (10, 50, 100, and 500 markers, respectively), and the repeatability of the STRUCTURE analysis of 50 randomly chosen markers was analyzed using CLUMPP.

## 5. Conclusions

Our study is a contribution to AEGIS and the work to avoid unwanted duplication holdings. We tested the SNP markers developed for *B. napus* and found that many of these genetic markers (nearly 6000) were suitable for an analysis of the genetic structure and duplicate identification of *B. oleracea*. Of these, a subset of 500 markers are recommended for a future large-scale analysis of *B. oleracea* var. *capitata*. Both the genetic SNP data and the morphological data demonstrate the complex relationships among old cabbage cultivars and show that similar accession names do not necessarily mean that accessions are genetically or morphologically similar. This emphasizes that in the case of old cultivars of cross-pollinating species such as cabbage, extra care should be taken when identifying duplicates.

## Figures and Tables

**Figure 1 plants-09-00925-f001:**
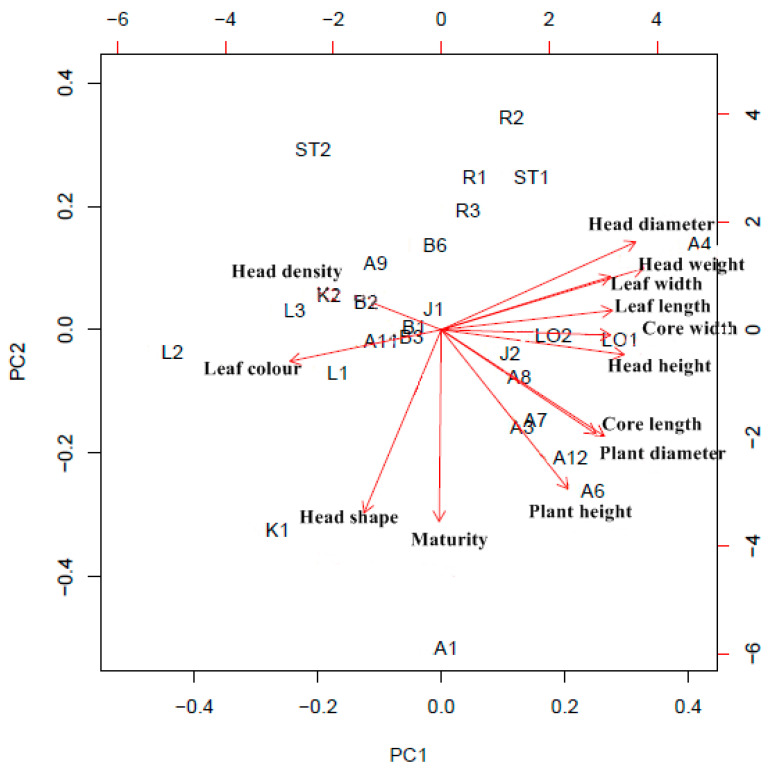
Morphological data: Principal component analysis (PCA) biplot where the descriptors of importance are given as arrows and where the length of an arrow is a measure of the descriptor’s variance and the angle between arrows is a measure of the correlation between descriptors, with a small angle expressing a high correlation. PC1 and PC2 explain 42% and 23% of the total variation. Accession names are abbreviated as codes; see Table 1.

**Figure 2 plants-09-00925-f002:**
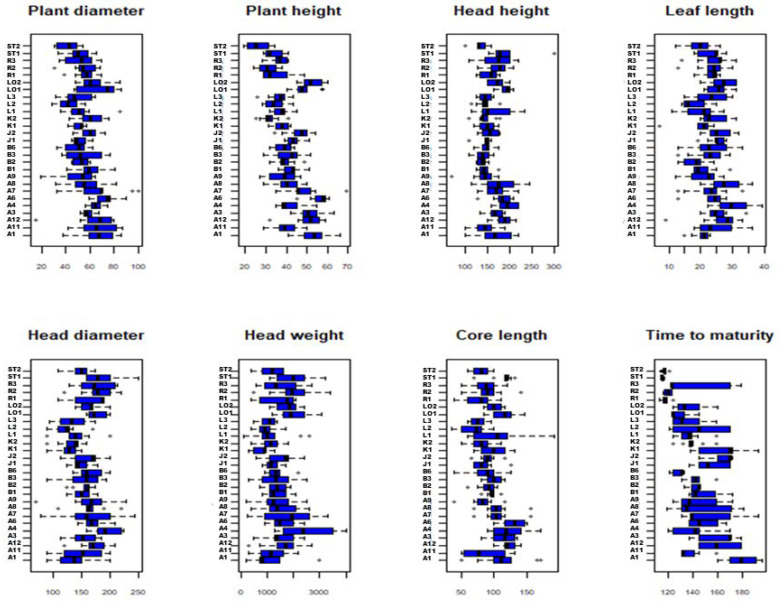
Boxplots describing the variation in the continuous morphological descriptors. Accession names are abbreviated as codes; see Table 1.

**Figure 3 plants-09-00925-f003:**
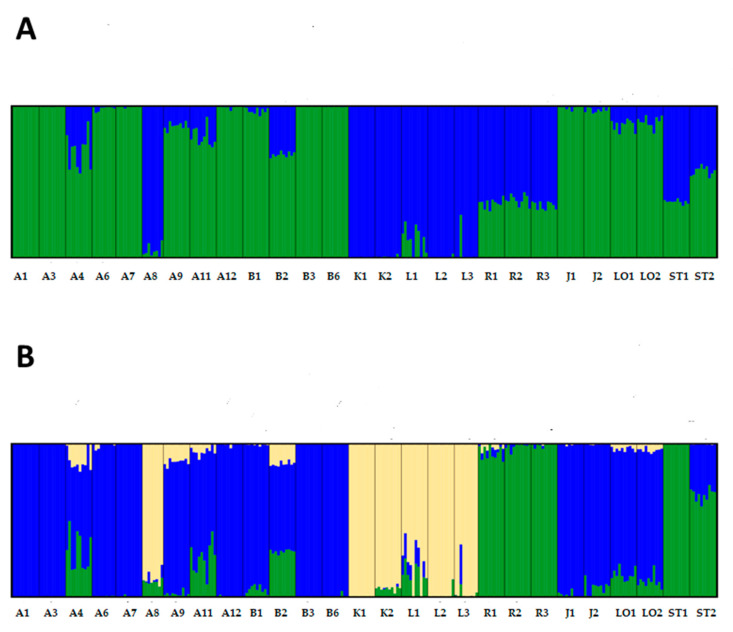
STRUCTURE analysis based on the full SNP dataset, assuming (**A**) two genetic clusters (K = 2); (**B**) three genetic clusters (K = 3). Different colours symbolize the different genetic clusters identified in each individual.

**Figure 4 plants-09-00925-f004:**
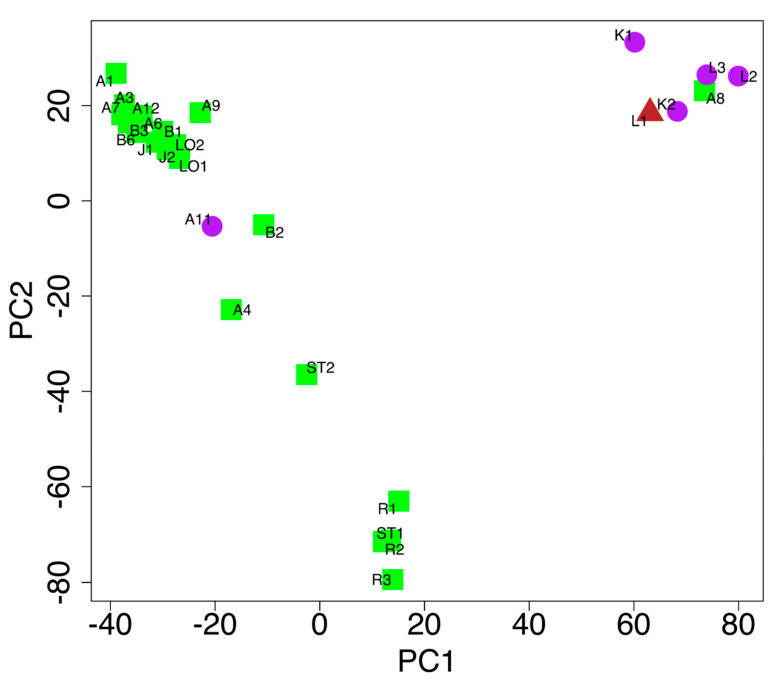
SNPs markers data: Accession-level principal component analysis (PCA) of the SNP markers, where the use of color refers to the leaf color of the cabbage accessions (purple circles = purple leaves, green squares = green leaves, reddish-brown triangles = purple/green leaves). PC1 and PC2 explain 15% and 10% of the total variation, respectively.

**Table 1 plants-09-00925-t001:** Overview of the accessions included in the study, sorted into groups based on accession name. For each accession, a code is given. Information is provided on the accession number, genebank holding institute, acquisition year, and donor institute. Acquisition year indicates the year when the accession entered the genebank holding institute, and the donor institute is the organization from which the material was sent.

Code	Accession Name	Accession Number	Gene Bank	Acquisition Year, Donor Institute
Amager Tall group				****
A1	Amager Hög	NGB11705	NGB	1995, Olson & Sons AB, Sweden
A3	Amager Høj Grøn Grami	NGB1875	NGB	1980, Dæhnfeldt A/S, Denmark
A4	Grami	K2537	VIR	1988, Unknown, Denmark
A6	Amager Høj, Grøn, Toftø 67	NGB1873	NGB	1980, FDB Frø, Denmark
A7	Amager Tall Resistent	K2475	VIR	1980, Unknown, Denmark
Amager Short pair				
A8	Amager Kurzstunkiger Orginal	K1485	VIR	1935, Ohlsens Enke, Denmark
A9	Amager L NF Orginal	K2248	VIR	1967, Unknown, Norway
Amager Winter pair				
A8	Amager Kurzstunkiger Orginal	K1485	VIR	1935, Ohlsens Enke, Denmark
A9	Amager L NF Orginal	K2248	VIR	1967, Unknown, Norway
Blåtopp group				
B1	Amager Faales Blatopp	K1181	VIR	1930, Norsk Frø, Norway
B2	Blatopp	K2250	VIR	1967, Unknown, Norway
B3	Blatopp Kvithamar	K2243	VIR	1967, Unknown, Norway
B6	Blåtopp Kvithamar	NGB4555	NGB	1984, Unknown, Norway
Kissendrup pair				
K1	Kissendrup	K111	VIR	1935, Unknown, Denmark
K2	Kissendrup Tagenshus	NGB1996	NGB	1980, Hansens Amagerfrø, Denamrk
Langendijker group				
L1	Langendijk Summer	K181	VIR	1959, Unknown, Denmark
L2	Langendijker Sommer Debut	NGB1997	NGB	1980, Ohlsens Enke, Denmark
L3	Langendijker Sommer Lanso	NGB1998	NGB	1980, Hansens Amagerfrø, Denmark
Ruhm v Enkhizen group				
R1	Ruhm von Enkhizen	NGB11718	NGB	1996, Hansens Amagerfrø, Denamrk
R2	Ruhm von Enkhizen Haba	NGB1888	NGB	1980, Hansens Amagerfrø, Denmark
R3	Ruhm von Enkhizen, B Hunderup	NGB2431	NGB	1982, Unknown, Denmark
Jåtunsalgets pair				
J1	Jatunsalgets Vinterkål Berbes	K2139	VIR	1959, Unknown, Norway
J2	Jåtunsalgets Vinterkål	NGB5007	NGB	1983, Unknown, Norway
Loke pair				
LO1	Loke	K2489	VIR	1982, Unknown, Sweden
LO2	Loke	NGB12050	NGB	1997, Unknown, Denmark
Stavanger Torv pair				
ST1	Stavanger Torv	K2175	VIR	1961, Unknown, Norway
ST2	Stavanger Torg	NGB8515	NGB	1990, NLH, Norway

**Table 2 plants-09-00925-t002:** Genetic diversity in individual accessions as measured by SNP markers.

Code	Average No Alleles	Nei′s h	Observed Heterozygosity
**A1**	1.1	0.06	0.06
**A3**	1.4	0.14	0.15
**A4**	1.8	0.27	0.30
**A6**	1.5	0.16	0.17
**A7**	1.3	0.14	0.12
**A8**	1.5	0.21	0.22
**A9**	1.3	0.13	0.14
**A11**	1.6	0.17	0.15
**A12**	1.4	0.14	0.15
**B1**	1.4	0.13	0.15
**B2**	1.4	0.14	0.18
**B3**	1.3	0.12	0.13
**B6**	1.4	0.13	0.14
**K1**	1.2	0.06	0.06
**K2**	1.4	0.14	0.15
**L1**	1.4	0.16	0.15
**L2**	1.4	0.13	0.10
**L3**	1.5	0.18	0.17
**R1**	1.5	0.17	0.19
**R2**	1.5	0.17	0.16
**R3**	1.4	0.14	0.13
**J1**	1.3	0.10	0.10
**J2**	1.4	0.14	0.15
**LO1**	1.6	0.19	0.22
**LO2**	1.6	0.19	0.18
**ST1**	1.3	0.11	0.14
**ST2**	1.3	0.12	0.13

**Table 3 plants-09-00925-t003:** Pairwise F_ST_ values within groups based on all markers (full dataset) and from subsets of 10 markers (subsample average). Subsample values that do not include the pairwise F_ST_ value for the full dataset are highlighted in bold.

Accession	Accession	Full Dataset	Subsample Average		
1	2	F_ST_	F_ST_	F_ST_ − 1 SE	F_ST_ + 1 SE
A1	A3	0.216	0.218	0.215	0.222
A1	A4	0.184	0.184	0.182	0.186
A1	A6	0.252	0.243	0.240	**0.247**
A1	A7	0.304	0.309	**0.305**	0.314
A3	A4	0.103	0.104	0.103	0.105
A3	A6	0.149	0.149	0.147	0.151
A3	A7	0.175	0.173	0.171	0.176
A4	A6	0.110	0.111	0.109	0.112
A4	A7	0.132	0.132	0.131	0.134
A6	A7	0.166	0.169	0.166	0.171
A8	A9	0.326	0.327	0.324	0.331
A11	A12	0.144	0.143	0.141	0.145
B1	B2	0.291	0.294	0.290	0.298
B1	B3	0.228	0.230	0.226	0.234
B1	B6	0.218	0.216	0.213	0.220
B2	B3	0.321	0.324	0.320	0.328
B2	B6	0.304	0.309	**0.305**	0.313
B3	B6	0.087	0.085	0.084	**0.087**
K1	K2	0.253	0.249	0.245	**0.253**
L1	L2	0.188	0.188	0.185	0.190
L1	L3	0.160	0.162	0.160	0.164
L3	L2	0.117	0.117	0.115	0.119
R1	R2	0.111	0.110	0.108	0.111
R1	R3	0.144	0.149	**0.147**	0.151
R2	R3	0.129	0.129	0.127	0.131
J1	J2	0.202	0.201	0.198	0.205
LO1	LO2	0.095	0.095	0.094	0.097
ST1	ST2	0.300	0.294	0.290	**0.298**

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
