# Peer review of "SNP Markers and Evaluation of Duplicate Holdings of Brassica oleracea in Two European Genebanks"

_plants, 2020, doi:10.3390/plants9080925_

Round 1

Reviewer 1 Report

The work of Palmé and co-authors reporting on the morphological and genetic characterization of Brassica oleracea is an excellent example of how joined-efforts between labs and countries are the best approach to maximize the potential of genetic resources. In their work, the authors have made available a collection of molecular markers and morphological descriptors that can be readily used for the characterization of the specimens available in other germplasm or seed banks. From the scientific point of view, this study is of interest at a global level, particularly in what concerns biodiversity conservation of crop species, under the context of climate change and population growth and their expected impact on agriculture. In my opinion, the MS deserves immediate acceptance, though I have suggested minor revisions.

Congratulations to both groups for this initiative.

Author Response

Reviewer 1

The work of Palmé and co-authors reporting on the morphological and genetic characterization of Brassica oleracea is an excellent example of how joined-efforts between labs and countries are the best approach to maximize the potential of genetic resources. In their work, the authors have made available a collection of molecular markers and morphological descriptors that can be readily used for the characterization of the specimens available in other germplasm or seed banks. From the scientific point of view, this study is of interest at a global level, particularly in what concerns biodiversity conservation of crop species, under the context of climate change and population growth and their expected impact on agriculture. In my opinion, the MS deserves immediate acceptance, though I have suggested minor revisions. Congratulations to both groups for this initiative.

Thanks for these good comments. We appreciate your detailed feedback (in the attached file).

We have corrected the suggested minor errors.

We included more references to discuss the results in section 3.2 and 3.3

We added reference to the morphological descriptors.

We added our work as a contribution to AEGIS in the conclusion.

Regarding other details, see our new manuscript with track changes. Due to comments from the other reviewers, we also changed other issues.

Reviewer 2 Report

Manuscript No. plants-854883

Title: Genetics and duplicate holdings of Brassica oleracea

Comment

In this study, the authors confirmed SNP marker set to evaluate the genetic duplication of B.oleracea. And then, using developed marker set and morphological traits, they evaluated B. oleraeca accessions with similar name. Identifying the duplicated plant accessions is important to manage the genebanks. However, the content is not clear and needs to rearrange.

1. Title is not suitable. I think the main content of this manuscript is to develop the marker set and to evaluate the duplicable B. oleracea.
2. Conservation of large genetic stock is one of the important object in genebanks. In this study, they mentioned “Examples from our dataset would be B3 and B6, or LO1 and LO2. However, there is no clear cut-off value to divide the accessions into similar and not similar and decisions regarding what to remove or not will, to some extent, be subjective.”
I agreed.
However, I think they should have mentioned an alternative to that in manuscript.

Author Response

Reviewer 2

In this study, the authors confirmed SNP marker set to evaluate the genetic duplication of B.oleracea. And then, using developed marker set and morphological traits, they evaluated B. oleraeca accessions with similar name. Identifying the duplicated plant accessions is important to manage the genebanks. However, the content is not clear and needs to rearrange.

We have made the introduction clearer, both regarding our mission and towards the content of the paper. As you suggested we have slightly changed the title. We also made extensive changes in the text (see details below and as track changes in the new version of the manuscript).

1.

Title is not suitable. I think the main content of this manuscript is to develop the marker set and to evaluate the duplicable B. oleracea.

We have changed the title to: “SNP markers and evaluation of duplicate holdings of Brassica oleracea”.

  1.  

Conservation of large genetic stock is one of the important object in genebanks. In this study, they mentioned “Examples from our dataset would be B3 and B6, or LO1 and LO2. However, there is no clear cut-off value to divide the accessions into similar and not similar and decisions regarding what to remove or not will, to some extent, be subjective.” I agreed. However, I think they should have mentioned an alternative to that in manuscript.

Regarding the mission of genebanks we agree that they should conserve large genetic stocks. However, our target was to identify unnecessary duplicate holdings. This to reduce the problems that many genebanks have with backlogs. And as it is an aim for the European genebank collaboration also. We think it has a value to identify duplicates and to remove these or at least not give priority to such duplicates. We rephrased the sentences from line 286 onwards. “Examples from our dataset that could be bulked, removed or given lower priority in the conservation could be B3 and B6, and one of the LO accessions. Our study has shown that using accession names alone is not a good strategy to reduce duplicate holdings, as the same or similar names does not mean identical genetic composition. A combined method, using both accession names and other passport data as a first step and then marker evaluations as a second step would be a better approach. Alternatively, morphological evaluations could be used or a more extensive passport data evaluation trying to trace transactions of accessions between genebanks, for example by using donor accession numbers or other relevant information. The ECPGR Brassica group has established an online tool for identifying duplicate holdings based on accession names and other passport data. This is a useful first step that could be taken into a next step with an extensive evaluation of the potential duplicates with the developed marker set.    

Reviewer 3 Report

Palmé et al. present here results from combining phenotypic and genotypic data in order to improve efficiency of Genebank conservation by identifying potential redundancy. The case study presented here is performed on various accession of Brassica oleraceae and take advantage of the relative homogeny of the Brassica complex to perform a polymorphism analysis. This study brings nicely to discussion the necessary conditions to identify “proper” duplicate that may need to be removed from collections as well as allowing identification of potentially miss-labelled accessions.

l 19. « strain » does not seem appropriate for vegetables.

L 24 “for future studies” to vague: for future Brassica oleracea genetic characterization? Overall, the abstract might not focus enough on the "larger picture": improve the efficiency of the strategy of conservation while proving genetic data can help in that.

L29. Any aim at trying generalising such approach to other species in genebanks? What to start with?

L486. capitata

Figure 2. Legend of the horizontal axis missing.

Table 1 might be easier to read as an histogram.

For the sake of curiosity: any of the polymorphisms observed in the L1 accession  could make sense in terms of leaf coloration?

Author Response

Reviewer 3

Palmé et al. present here results from combining phenotypic and genotypic data in order to improve efficiency of Genebank conservation by identifying potential redundancy. The case study presented here is performed on various accession of Brassica oleraceae and take advantage of the relative homogeny of the Brassica complex to perform a polymorphism analysis. This study brings nicely to discussion the necessary conditions to identify “proper” duplicate that may need to be removed from collections as well as allowing identification of potentially miss-labelled accessions.

Thanks for your comments. As you will see, we have submitted a new version of the manuscript. Due to comments from some of the other reviewers, we changed some of the text, and all changes are marked as Track Changes. We also included your points as suggested below:

L 19. « strain » does not seem appropriate for vegetables.

We changed to “selections” at relevant places in the document. In the abstract we instead of: “In the past, and especially in vegetables, different selections or strains within the same varieties were common…”, we changed to “In the past, and especially in vegetables, different selections within the same varieties were common…”

L 24 “for future studies” to vague: for future Brassica oleracea genetic characterization? Overall, the abstract might not focus enough on the "larger picture": improve the efficiency of the strategy of conservation while proving genetic data can help in that.

We also changed the start of the abstract and included a sentence about the larger picture:  “Around the world there are more than 1500 genebanks storing plant genetic resources to be used in breeding and research. Such resources are essential for future food security but many genebanks experience backlogs in their conservation work, often combined with low budgets. Therefore, avoiding duplicate holdings is on the agenda”.    

We changed the sentence in the old line 24 to: “…. and a subset of 500 SNP markers are suggested for or future genetic characterization in Brassica oleracea”.

L29. Any aim at trying generalising such approach to other species in genebanks? What to start with?

We added in line 29-30: “…old cultivars of cross-pollinated species such as cabbage. We think the same approach can be applied to other Brassica vegetables, carrot, onion, leek and others.”  

L486. Capitate

We changed

Figure 2. Legend of the horizontal axis missing.

We have changed

Table 1 might be easier to read as an histogram.

Table 1 as a histogram is hard to make, as it would overlap with Table S1, so we kept Table S1 unchanged. 

For the sake of curiosity: any of the polymorphisms observed in the L1 accession could make sense in terms of leaf coloration?

We now added a sentence in the discussion about this “The polymorphism observed in accession L1 ( compared to L2 and L3, see Figure 3) could relate to differences with both green and purple leaves in this accession”. 

Reviewer 4 Report

Dear Editor,

The manuscript plants-854883 has  some major errors in the designing of the experiments and analyzing the data. Therefore, it has to be revised at major level. The reviewed comments are attached below- 

Ms_Plant-854883

Overall Comments: In the vegetable research, the manuscript would be a good source of information for harnessing the genetic diversity in the breeding program. However, The manuscript plants-854883 has some major errors in the designing of the experiments and analyzing the data. Therefore, it must be revised at major level.

Comments-

  1. Introduction is insufficient with background information. It needs to be updated with recent information on germplasm purification in the germplasm banks.
  2. In the plant material and cultivation, what was the experimental design and model to control mechanical and human errors in the experiments?
  3. How much diverse are the accessions based on the previous studies?
  4. What was the criterion for choosing two groups of cabbage accessions (92 and 178)?
  5. Why were 10 individuals per accession use for genotyping analysis? Was there any DNA pool or something else?
  6. Why did you merge two data sets? Were they two different experiments or were these two experiments conducted on time? Explain?
  7. In the materials and methods, what morphological traits were measured not mentioned and they must be explained with the procedure and protocols?
  8. In the results, PCA plots should have shown explained variation in PC1 and PC2. Why did author show two figures with the same PC1 and PC2 and it should be at least PC1, PC2, and PC3.
  9. Figure quality and resolution is very poor and not readable. The figures must be improved.

Thanks

Author Response

Reviewer 4

Overall Comments: In the vegetable research, the manuscript would be a good source of information for harnessing the genetic diversity in the breeding program. However, The manuscript plants-854883 has some major errors in the designing of the experiments and analyzing the data. Therefore, it must be revised at major level.

Thanks for your review and useful suggestions for improving the quality of our manuscript. We have critically examined your points and have changed parts of our manuscript accordingly. See detailed response to each of your comments below. We have included a new version with changes marked as Track Changes.

Comments

1.

Introduction is insufficient with background information. It needs to be updated with recent information on germplasm purification in the germplasm banks.

We have updated the introduction with recent information on the issue you bring up. Among the new references we added were:

Fu, Y.‐B. (2017), The Vulnerability of Plant Genetic Resources Conserved Ex Situ. Crop Science, 57: 2314-2328. doi:10.2135/cropsci2017.01.0014

Alercia, A., López, F.M., Sackville Hamilton, N.R. and Marsella, M., 2018. Digital Object Identifiers for food crops - Descriptors and guidelines of the Global Information System. Rome, FAO.

Zamir D (2013) Where Have All the Crop Phenotypes Gone? PLoS Biol 11(6): e1001595. https://doi.org/10.1371/journal.pbio.1001595

McCouch, S. R., McNally, K. L., Wang, W. & Sackville Hamilton, R. Genomics of gene banks: A case study in rice. Am J Bot 99, 407–423, https://doi.org/10.3732/ajb.1100385 (2012).

Poland, J. A. & Rife, T. W. Genotyping-by-Sequencing for Plant Breeding and Genetics. Plant Genome-Us 5, 92–102, https://doi.org/10.3835/plantgenome2012.05.0005 (2012).

Huang, Y. F., Poland, J. A., Wight, C. P., Jackson, E. W. & Tinker, N. A. Using genotyping-by-sequencing (GBS) for genomic discovery in cultivated oat. PloS One 9, e102448, https://doi.org/10.1371/journal.pone.0102448 (2014).

Noelle L. Anglin, Ahmed Amri, Zakaria Kehel, and Dave Ellis. Biopreservation and Biobanking. Oct 2018.337-349.http://doi.org/10.1089/bio.2018.0033

Singh, N., Wu, S., Raupp, W.J. et al. Efficient curation of genebanks using next generation sequencing reveals substantial duplication of germplasm accessions. Sci Rep 9, 650 (2019). https://doi.org/10.1038/s41598-018-37269-0

McCouchS., BauteG.J. and BradeenJ. et al. 2013. Agriculture: Feeding the future. Nature 499:23–24. doi:10.1038/499023a

van TreurenR., van HintumT.J.L. 2014. Next‐generation genebanking: Plant genetic resources management and utilization in the sequencing era. Plant Genet. Resour. 12:298–307.

2.

In the plant material and cultivation, what was the experimental design and model to control mechanical and human errors in the experiments?

In M&M we included more details:

“ Plants were randomized within each pair/triplets and with single plant scoring of 10 plants per accession. Planting distance was 50 cm between the plants. The work was done at Alnarp, Sweden (55°’N, 13°’E), the soil was loamy clay and fertilization was 100 kg ha-1 PROMAGNA 11-5-18™ (Yara, Norway) at planting and 30 kg ha-1 YaraMila 22-0-12™ (Yara, Norway) one month after planting. We irrigated and applied biological control and fungicides.”

3.

How much diverse are the accessions based on the previous studies?

We added a sentence in the introduction of section 4.2: “As far as we know, these accessions have not previously been included in any molecular studies. DNA extraction was done on the same ten plants per accession and from the same experiment that were morphologically characterized.” 

4.

What was the criterion for choosing two groups of cabbage accessions (92 and 178)?

The first run with 92 individuals was a test run and since that was successful, the second run with 178 individuals was done in the same way to increase the number of individuals analysed. The number of individuals (92 and 178) were chosen for technical reasons to fit the SNP array analysis.

5.

Why were 10 individuals per accession use for genotyping analysis? Was there any DNA pool or something else?

We have explained this now: “Since cabbage accessions are expected to harbour substantial within-accession variation, many individuals per accession needed to be investigated. By analysing 10 individuals per accessions we gain an adequate picture of the within species variation, but are still able to include many accessions in the study.” 

Furthermore, we have added the following to row 447: “… , in order to capture the within accession variation.”

6.

Why did you merge two data sets? Were they two different experiments or were these two experiments conducted on time? Explain?

This is linked to what was explained above in feedback point 4. We changed the words “two dataset” to “two runs” and gave the explanation as described:

The first run with 92 individuals was a test run and since that was successful, the second run with 178 individuals was done in the same way to increase the number of individuals analyzed. The number of individuals (92 and 178) were chosen for technical reasons to fit the SNP array.

After merging of the two SNP runs, failed and invariant markers were….

7.

In the materials and methods, what morphological traits were measured not mentioned and they must be explained with the procedure and protocols?

We added: “We evaluated the plants when ready for harvesting. We applying SI units for plant, lead, head and core size parameters and UPOV (2004) descriptors for leaf colour, head shape and head density. Details are provided in Table S1.”

A new reference was added: UPOV (2004). UPOV guidelines for the conduct of tests for distinctness, uniformity and stability TG/48/7. Geneva: International union for the protection of new varieties of plants.

8.

In the results, PCA plots should have shown explained variation in PC1 and PC2. Why did author show two figures with the same PC1 and PC2 and it should be at least PC1, PC2, and PC3.

We now have specified in the Figure captions of the two relevant figures (Figure 1 and Figure 4) that these are based on morphological data and SNPs marker data, respectively.  For the morphological PCA, PC1 and PC2 explain 42% and 23% of the total variation and this is now stated in the figure caption 1. For the SNP markers PCA, PC1 and PC2 explain 15% and 10% of the variation and this is added to the figure caption 4.

 The figures of PCA plots show the results of analysis of different data or different analysis:

  • Figure 1 depicts the PCA for the morphological descriptors
  • Figure 4 depicts the accession level PCA of the SNP markers
  • Figure S2 depicts the individual level PCA of the SNP markers

The figure texts describe the differences of the PCAs

9.

Figure quality and resolution is very poor and not readable. The figures must be improved.

We have improved the resolution of Figure 2, to make the details more visible.

We have improved the resolution in figure 3a and 3b. They are now attached as separate files, and we would like to ask the production team at MDPI to merge them into one figure. Regarding the supplementary figures, we have improved the resolution of figure S2 and Figure S4, and we have removed unnecessary white empty space in Figure S5 and S6.

We improved the quality of Figure 1

We kept Figure 4 unchanged. We also think Figure S1 and Figure S3 look good and with a satisfactory resolution.

We have improve the quality of Figure 2. Figure 3a and b we have in good resolution so they should be fine.

We have improved the resolution of Figures S2 and Figure S4.

Round 2

Reviewer 2 Report

Thank you for your sufficient reply and revision.

Author Response

Thank you for your work in reviewing our manuscript. We have now updated a new version with track changes.

Reviewer 4 Report

Overall Comments: In the vegetable research, the manuscript would be a good source of information for harnessing the genetic diversity in the breeding program. Therefore, the manuscript plants-854883 has been revised and has addressed most of the major errors.

Comments-

  1. The MS still has a few minor typos or errors that should be addressed before accepting the MS.
  2. Experimental design has not been still explained in the MS.
  3. Figure-3, the image quality is still poor, which has to be improved with better resolution.

Author Response

The MS still has a few minor typos or errors that should be addressed before accepting the MS.

We put caption letters to each word of importance in the title and updated the title according to editor’s note.

Line 41, we removed the double comma.

Line 43, we added “and maintaining” to the sentence to specify what duplication is.

Line 50, we changed “at” to “in”

Line 72, we changed “paper” to “study” and added a comma.

Line 73, we used semi-colon instead of colon.

Line 111-114, we improved the language in the caption text by five minor changes.

Line 125, we added a “e”

Line 175-176, the sentence structure is improved (as it gave no meaning as it stood).

Line 163, we added “s” in “accessions”.

Line 164, we changes “were” to “was”.

Line 180-1811, we moved “all” to the front of the sentence and added the word “same” to the sentence to improve the meaning of the sentence.

Line 188-189, we improved the sentence structure.

Line 194, we changed “extent” to “degree”.

Line 196, we changed “centre” to “center”.

Line 199-201, we changed the sentence structure.

Line 206-209, changed from purple or green-headed to purple or green-leafed throughout the sentence, as we talk about leaf colour in the following sentence and in the descriptor list.

Line 224, we deleted “however” in this sentence.

Line 226, we change the grammar of the sentence so that it became correct.

Tine 233, we changed from “was” to “were”.

Line 238, we added a comma.

Line 249-256, we highlighted the genomes with tag-marks ‘A’ genome and ‘C’ genome, respectively. We changed the structure of the sentence in line 251.

Line 290, we added a comma.

Line 367, we changed the place of the “n” in “Scandinavian”.

Line 383-384, we changed the structure of the sentence.

Line 409, we added a “w” In “shown”.

Line 412, we simplified the sentence.

Line 415, we deleted this sentence.

Line 423, we removed a “s”.

Line 434, we improved the sentence.

Line 447-449, we changed the text (see comment below on experimental design).

Line 459-461, we made the sentence more clear and split it into two sentences.

Line 466; we changed from “whereafter” to “hereafter”.

Line 477, we added a “–“, so it now says: “re-suspended”.

Line 481, we added “and” into the sentence.

Experimental design has not been still explained in the MS.

We added the design into M&M section and Line 446-451 we changed the text to:

“In this study, we examined another ten groups (pair/triplets) as demonstrated in Table 1 to see if there were differences between accessions within a group.  For each group we planted ten plants per accession. These plants were randomized and we characterized each of the ten plants. The same was done for the following group, so the study consisted of ten completely randomized sub-experiments.  ”

For the statistics, we clarified by saying: “ANOVA was done for each numeric character and included data from all individuals in that group. If the ANOVA indicated significant differences, a Tukey multiple comparison of means [48] was used to identify accessions that differed from each other. Chi-square statistics was used for categorical characters.”

Figure 3, the image quality is still poor, which has to be improved with better resolution.

We have simplified and  improved the resolution.